# Likelihood-based fine-tuning of protein language models for few-shot fitness prediction and design

**Alex Hawkins-Hooker** [1 2 *]  **Jakub Kmec** [2]  **Oliver Bent** [2]  **Paul Duckworth** [2]

## Abstract

Although various schemes have been proposed for exploiting the distributional knowledge captured by protein language models (PLMs) to enhance supervised fitness prediction and design, lack of head-to-head comparison across different prediction strategies and different classes of PLM has made it challenging to identify the best-performing methods, and to understand the factors contributing to performance. Here, we extend previously proposed ranking-based loss functions to adapt the likelihoods of family-based and masked protein language models, and demonstrate that the best configurations outperform state-of-the-art approaches based on frozen embeddings in the low-data setting. Furthermore, we propose ensembling strategies that exploit the strong dependence of the mutational distributions learned by PLMs on sequence context, showing that they can be used to guide efficient optimisation strategies over fitness landscapes.

## 1. Introduction

Protein language models (PLMs) fit to the distribution of natural sequences learn to implicitly model functional and structural constraints relevant to protein function, with their likelihoods forming effective zero-shot predictors of the fitness effects of mutations (Meier et al., 2021; Notin et al., 2022). In practical protein design scenarios, it is often possible to use experimental techniques to generate labelled datasets associating sets of sequences with quantitative measurements of biological properties of interest, however experimental constraints mean that it might only be feasible to generate measurements for tens or hundreds of proteins

at a time (Biswas et al., 2021). It is therefore of considerable interest to ask how the zero-shot prediction capacities of PLMs can be combined with small labelled datasets to achieve improved predictive performance.

One popular paradigm for exploiting the information in pretrained PLMs involves extracting sequence representations and feeding these as inputs into task-specific downstream predictive models (Alley et al., 2019; Biswas et al., 2021; Rao et al., 2019; Dallago et al., 2021; Notin et al., 2023b). However, recent trends in natural language processing have shown the benefits of directly adapting the distributions of models using task-specific labelled or preference data (Ouyang et al., 2022; Rafailov et al., 2023), thereby fully exploiting the distributional knowledge contained in the original pretrained model. Although related fine-tuning strategies have been considered in the context of fitness prediction with unconditional autoregressive PLMs (Krause et al., 2021), previous work has not addressed whether similar strategies can effectively be applied across different classes of PLM, which often outperform unconditional autoregressive models for fitness prediction (Notin et al., 2023a), nor how to exploit fine-tuning to improve performance in uncertainty-guided design tasks. Moreover, there has been relatively limited direct comparison of these fine-tuning strategies to alternative PLM-based fitness prediction strategies, including recent innovations in architectures for operating over frozen PLM embeddings, (Notin et al., 2023b), making it difficult to assess their utility in practice.

Seeking to address this gap, in this paper we (i) show that ranking losses can be extended to adapt the likelihoods of leading zero-shot fitness predictors trained with both masked and family-based autoregressive language modelling objectives, (ii) provide direct comparison with state-of-the-art approaches based on frozen protein language model embeddings (Notin et al., 2023b), as well as fine-tuning with added regression heads, thereby offering compelling empirical evidence for the effectiveness of the proposed fine-tuning schemes, and (iii) develop ensembling strategies compatible with these fine-tuning schemes, demonstrating their effectiveness in both supervised and multi-round design settings.

---

[*]Work completed during an internship at InstaDeep. [1]AI Centre, University College London, London, United Kingdom [2]InstaDeep Ltd.. Correspondence to: Alex Hawkins-Hooker <ucabawk@ucl.ac.uk>, Paul Duckworth <p.duckworth@instadeep.com>.

*Accepted at the 1st Machine Learning for Life and Material Sciences Workshop at ICML 2024.* Copyright 2024 by the author(s).

## 2. Background

Two recent works have advocated the use of ranking-based loss functions (Krause et al., 2021; Brookes et al., 2023). In particular, they suggest parameterising a Bradley-Terry model (Bradley & Terry, 1952) with a learned function of the sequence. The Bradley-Terry model represents the probability that a given sequence $x_i$ has higher fitness $y$ than another sequence $x_j$ by parameterising a binary classifier via the difference in scores of each sequence under a learned scoring function $s_\theta(x)$:

$$p(y(x_i) > y(x_j)) = \sigma(s_\theta(x_i) - s_\theta(x_j)), \qquad (1)$$

where $\sigma$ is the logistic sigmoid function. The model can be fit to data by maximising the likelihood of the complete set of pairwise comparisons between the fitness values of sequences with respect to the parameters $\theta$ of the scoring function. Concretely, in our implementation, given a batch of $B$ sequences $x_1, ..., x_B$, the resulting loss is:

$$\mathcal{L} = \sum_{i=1}^{B} \sum_{j=1}^{B} -\mathbb{I}(y(x_i) > y(x_j)) \log \sigma(s_\theta(x_i) - s_\theta(x_j)), \tag{2}$$

where $\mathbb{I}$ is an indicator function. In this way, fitness prediction for a dataset of size $N$ is converted from a regression problem with $N$ labels into a binary classification problem with $N \times N$ labels.

### 2.1. Ranking-based fine-tuning of autoregressive protein language models

To use the Bradley-Terry model to fine-tune an autoregressive protein language model, Krause et al. (2021) propose using an unconditional sequence log-likelihood as the scoring function:

$$s_\theta(x) = \sum_{i=1}^{L} \log p(x_i|x_{<i}). \tag{3}$$

Since the log-likelihoods of pretrained autoregressive protein language models are reasonably well-correlated with the fitness effects of mutations (Notin et al., 2022), the difference in log-likelihoods used to parameterise the Bradley-Terry model of Equation 1 can already produce an effective pairwise classifier at initialisation, which may help maximise the effectiveness of fine-tuning on small datasets.

## 3. Likelihood-based fine-tuning of masked and family-based protein language models

Unconditional autoregressive models often underperform other classes of model including conditional autoregressive models and masked language models in fitness prediction settings (Notin et al., 2023a). We therefore extend fine-tuning via the Bradley-Terry model to accommodate these

more performant PLMs. To do so, we incorporate the additional conditioning information $c$ exploited by these models into conditional scoring functions $s_\theta(x, c)$:

$$p((y(x_i) > y(x_j))|c) = \sigma(s_\theta(x_i, c) - s_\theta(x_j, c)). \tag{4}$$

Below, we will consider cases where $c$ represents either a wild-type sequence or a multiple sequence alignment (MSA), since conditioning on evolutionary context is especially effective in fitness prediction (Truong Jr & Bepler, 2023), but note that the same approach could be applied to models which condition on protein structure (Hsu et al., 2022b).

### 3.1. Masked protein language models

Masked language models do not define a sequence-level likelihood that can directly be used as a scoring function. Instead we build on the zero-shot scoring strategies proposed by Meier et al. (2021) to allow these models to be fine-tuned with ranking-based losses, similar to other concurrent work (Zhao et al., 2024). Concretely, we utilize the 'wild-type marginals' scoring function from Meier et al. (2021). Under this strategy the score for a mutated sequence is given by the summation of the log-likelihood ratios between mutated and wild-type amino acids across mutated positions, given the unmasked wild-type sequence as input:

$$s_\theta(x, x^{\text{wt}}) = \sum_{i:x_i^{\text{wt}} \neq x_i} \log p(x_i|x^{\text{wt}}) - \log p(x_i^{\text{wt}}|x^{\text{wt}}). \tag{5}$$

Since all sequences are scored under the residue distributions obtained by feeding the wild-type sequence through the model, a set of mutated sequences of arbitrary size can be scored using a single forward pass, making both fine-tuning and prediction extremely efficient.

### 3.2. Family-based protein language models

Family-based protein language models represent the conditional distribution over family members given a subset of other family members (Rao et al., 2021; Hawkins-Hooker et al., 2021; Ram & Bepler, 2022; Truong Jr & Bepler, 2023). These models have proved especially effective as zero-shot fitness predictors, due to their ability to explicitly condition on evolutionary context to predict the effects of mutations.

In this paper we work with PoET (Truong Jr & Bepler, 2023), which models entire protein families autoregressively. To produce zero-shot predictions given a mutant sequence $x$ and an MSA $M = \{m^{(1)}, ..., m^{(N)}\}$ of homologues of a wild-type sequence $x^{\text{wt}}$, PoET computes the likelihood of the mutant $x$ given the MSA. To exploit this capacity to condition on family members during fine-tuning, we condition the autoregressive scoring function in Equation 3 on

the sequences in the MSA:

$$s_\theta(x, M) = \sum_{i=1}^{L} \log p(x_i | x_{<i}, M).$$ (6)

Since PoET operates natively on unaligned sequences and is sensitive to alignment depth, we subsample a small set of sequences from the MSA and discard gaps before feeding them into the model, following (Truong Jr & Bepler, 2023). To increase the efficiency of fine-tuning PoET, in practice we cache a single set of hidden layer representations obtained by passing the subsampled MSA $M$ through the model, and fine-tune only the mapping between these frozen representations and the sequence likelihoods, decoupling the encoding of prior context from the decoding of future amino acids given this context (Appendix C).

### 3.3. Uncertainty quantification with evolutionary context ensembles

The amino acid output distributions learned by protein language models depend heavily on sequence context. We propose to exploit this property to build ensembles of fine-tuned PLMs, in which each ensemble member sees a different, but approximately biologically equivalent, context. Concretely, to fine-tune an ensemble of PoET models, for each fitness dataset we sub-sample a set of $K$ input MSAs $M_k$ from the full MSA associated with the wild-type sequence. We then fine-tune a separate set of parameters to minimise the ranking loss conditioned on each MSA, producing $K$ sets of parameters, each specialised to a single input MSA. To score sequences, we use an ensembled scoring function:

$$s_{\theta_1, ..., \theta_K}(x, M) = \frac{1}{K} \sum_{k=1}^{K} s_{\theta_k}(x, M_k).$$ (7)

To achieve a similar effect with ESM-1v, which does not use MSAs, we instead sample a set of $K$ input masks, and fine-tune a separate set of parameters for each input mask, exploiting the intuition that differently masked sequences are functionally equivalent, but may nonetheless produce different outputs when passed through the model.

### 3.4. Relationship to preference learning strategies for LLMs

Direct preference optimisation (DPO) (Rafailov et al., 2023) is a recently proposed method for aligning large language models (LLMs) using datasets of human preference data. DPO also uses scoring functions from pretrained models to parameterise a Bradley-Terry model. Instead of parameterising a classifier directly via differences in log likelihoods, DPO uses the difference in scaled log likelihood *ratios* between the fine-tuned model and a frozen reference model. Thus the probabillity that a completion $x_1$ is preferred to a completion $x_2$ given an input prompt $c$ is modelled as:

$$p_\theta(x_1 \succ x_2 | c) = \sigma\left( \beta \log \frac{p_\theta(x_1|c)}{p_{\text{ref}}(x_1|c)} - \beta \log \frac{p_\theta(x_2|c)}{p_{\text{ref}}(x_2|c)} \right).$$ (8)

In our notation, the DPO preference model therefore amounts to a particular choice of scoring function $s_\theta(x, c) = \beta \log \frac{p(x|c)}{p_{\text{ref}}(x|c)}$. Assuming an autoregressive decomposition of $p(x|c)$, this scoring function is equivalent to the conditional autoregressive scoring function in Equation 6 if the reference model is chosen to be constant and $\beta = 1$.

The non-constant reference model in DPO imposes a KL penalty on the deviation between the fine-tuned $p_\theta$ and the reference model, which helps prevent collapse in the fine-tuned distribution (Rafailov et al., 2023). Although some recent work has reported success in adapting DPO to the protein fitness prediction setting (Lee et al., 2023), in our own experiments we did not find this regularisation necessary to achieve good performance, possibly owing to the fact that we require neither generations from the model nor generalisation to different 'conditions' at test time, unlike typical applications of DPO.

## 4. Few-shot fitness prediction

We study the performance of fitness prediction strategies on mutational landscapes from ProteinGym (Notin et al., 2023a). We utilise two subsets of ProteinGym: the validation set of 8 representative single-mutant landscapes selected by Notin et al. (2023b), and a set of multi-mutant landscapes, chosen to constitute a non-redundant set of the most diverse landscapes available (Appendix B).

In contrast to some prior work (e.g. Notin et al. (2023b)), we focus explicitly on the low-data setting. For each landscape, we train all methods on $n = 128$ or $n = 512$ sequences randomly sampled from the landscape and evaluate on either 2000 (for single-mutant landscapes) or 5000 (for multiple-mutant landscapes) randomly sampled held-out sequences. An additional set of 128 randomly sampled sequences is used as a validation set to perform early stopping. For each landscape, and each $n$, we generate three sets of random splits, and report test set Spearman correlation averaged across the three splits. For models trained with ranking losses, the Spearman correlation is computed between the scoring function $s_\theta(x, c)$ and the ground truth fitness values.

### 4.1. Fitness prediction strategies

We evaluate the performance of the likelihood-based fine-tuning strategies introduced in Section 3 on the selected landscapes. To attain an understanding of the effectiveness of these strategies across different classes of PLM, we apply

them to the masked language model ESM-1v (Meier et al., 2021), the unconditional autoregressive model ProGen2 (Nijkamp et al., 2023), and the family-based autoregressive model PoET (Truong Jr & Bepler, 2023). For ProGen2 we obtained slightly better results with the 'small' checkpoint model than the 'medium' one, so report the former. In each case, the model is fine-tuned by parameterising the Bradley-Terry model of Equation 1 via the corresponding scoring functions in Sections 2 and 3.

We compare to two sets of baselines, representative of widely used approaches that either (i) fine-tune PLMs by adding a regression head (Rao et al., 2019), or (ii) train new models on top of frozen language model embeddings (Notin et al., 2023b). In the first case, we add linear regression heads to both ESM-1v and PoET, and fine-tune all parameters. As the leading example of the second class of approaches, we compare against ProteinNPT (Notin et al., 2023b), a state-of-the-art model operating on top of frozen language model embeddings. As additional baselines, we include the 'augmented density' strategies used by (Notin et al., 2023b). These models are regression models taking as input the zero-shot predictions of a PLM as well as either a one-hot representation of the mutated sequence (Hsu et al., 2022a), or an embedding extracted from a PLM (Notin et al., 2023b). We refer to these distinct choices of augmented density representation as 'OHE augmented' (OHE aug.) and 'Embedding augmented' (Emb. aug.) respectively, following Notin et al. (2023b).

Hyperparameters for fine-tuned models are selected based on performance on the single mutant set, consistent with the practice used for ProteinNPT and associated baselines. We report metrics obtained when using these hyperparameters on both single-mutant and multiple mutant landscapes for each method. Additional descriptions of models and training details are provided in Appendix D.

## 4.2. Results

**Ranking-based fine-tuning outperforms regression-based fine-tuning**  We first focus on the comparison between ranking-based fine-tuning and regression-based fine-tuning, *using the same models*. For PoET, ranking-based fine-tuning performs best across both dataset sizes for single and multi-mutant landscapes (Table 1). Regression-based fine-tuning is nonetheless a strong baseline, performing slightly better than the best ProteinNPT configuration. For ESM-1v, ranking-based fine-tuning performs much better than regression-based fine-tuning on the single mutant landscapes, but worse on the $n = 512$ multi-mutant landscapes. Unlike regression-based fine-tuning via a linear head, the wild-type marginals scoring rule used in ranking-based fine-tuning of ESM-1v is unable to capture the interactions between multiple mutations, since it assumes that mutation

effects are additive. In contrast, the scoring functions used for ranking-based fine-tuning of autoregressive models are able to capture interactions between mutations and therefore perform well on the multiple mutants datasets. Indeed the unconditional autoregressive ProGen models outperform the ranking-based version of ESM-1v on the multiples datasets, but not on the single mutant datasets, while the family-based autoregressive model PoET achieves strong performance across the board. We note that these observations do not necessarily point to a fundamental limitation of fine-tuning masked PLMs with ranking-based losses; more expressive masked PLM scoring functions involving multiple forward passes may be able to achieve better performance at the price of increased computational expense.

**Ranking-based fine-tuning outperforms models trained on frozen embeddings**  We next focus on the comparison between the best-performing ranking-based fine-tuning schemes and baselines relying on frozen embeddings. Ranking-based fine-tuning of PoET outperforms Protein-NPT across all settings (Table 1), with the gap largest in the $n = 128$ regime, suggesting that directly adapting the likelihoods of the pretrained model is especially helpful for maximising performance given very limited data. Notably this is not simply by virtue of PoET producing better zero-shot predictions: on the single mutant datasets, the zero-shot ESM-1v predictions used by ProteinNPT (ESM-1v) outperform those produced by PoET. ESM-1v fine-tuned with a ranking loss is also slightly better than ProteinNPT on the single-mutant datasets, but performs worse on the multi-mutant datasets, for the reasons discussed above. For both ESM-1v and PoET, the proposed ensembling strategies further improve performance, sometimes substantially, and show improved uncertainty calibration, as measured by the negative log likelihood of pairwise classifications from the test set (Table 3).

**Ranking-based fine-tuning generalises to unseen positions**  Random splits provide an estimate of performance on heldout data. However, similar mutations can occur in both train and test sets (e.g. related amino acid substitutions at the same position), meaning that measuring performance on predicting the effects of these mutations does not necessarily test a model's capacity for generalisation (Notin et al., 2023b). We assess the capacity of models to generalise to mutations at unseen positions in the single mutant datasets by reporting the performance of all models for mutations in the $n = 128$ test sets occurring at positions at which no mutations were present in the training set sequences (Table 2). While there is a clear drop in performance at these unseen positions, PoET fine-tuned with a ranking loss still performs the best, indicating that it is able to generalise across positions better than other methods.

*Table 1.* Spearman correlation on 8 single mutant landscapes and 5 multiple mutant landscapes from ProteinGym. Results for $n = 0$ are computed on the $n = 128$ test splits. Where methods use a frozen base model to produce embeddings and zero-shot predictions, the base model type is provided in parentheses, and zero-shot performance is that of the base model.

| Model name (base model) | Loss type | Singles | | | Multiples | | |
|---|---|---|---|---|---|---|---|
| | | $n = 0$ | $n = 128$ | $n = 512$ | $n = 0$ | $n = 128$ | $n = 512$ |
| ESM-1v | ranking | 0.384 | 0.552 | 0.637 | 0.425 | 0.653 | 0.736 |
| ESM-1v + linear head | regression | - | 0.425 | 0.583 | - | 0.649 | 0.780 |
| PoET | ranking | 0.417 | **0.589** | **0.668** | 0.592 | **0.738** | **0.806** |
| PoET + linear head | regression | - | 0.554 | 0.649 | - | 0.711 | 0.784 |
| ProGen2 small | ranking | 0.385 | 0.521 | 0.623 | 0.358 | 0.670 | 0.768 |
| ProteinNPT (MSAT) | regression | 0.399 | 0.545 | 0.635 | 0.534 | 0.689 | 0.782 |
| ProteinNPT (ESM-1v) | regression | **0.437** | 0.497 | 0.602 | 0.392 | 0.646 | 0.775 |
| Emb. aug. (MSAT) | regression | 0.399 | 0.541 | 0.627 | 0.534 | 0.707 | 0.783 |
| Emb. aug. (ESM1v) | regression | **0.437** | 0.532 | 0.609 | 0.392 | 0.638 | 0.765 |
| OHE aug. (MSAT) | regression | 0.399 | 0.465 | 0.500 | 0.534 | 0.643 | 0.745 |
| OHE aug. (ESM1v) | regression | **0.437** | 0.496 | 0.531 | 0.392 | 0.556 | 0.709 |
| OHE | regression | - | 0.303 | 0.494 | - | 0.467 | 0.671 |

*Table 2.* Single-mutant Spearman correlations for test set mutations at seen and unseen positions (n=128). Test set mutants are assigned to the unseen set if they contain mutations in sequence positions at which none of the training set sequences have mutations.

| Model name | Loss type | Spearman | |
|---|---|---|---|
| | | Seen | Unseen |
| ESM1v | ranking | 0.587 | 0.474 |
| ESM1v + linear head | regression | 0.484 | 0.303 |
| PoET | ranking | **0.617** | **0.531** |
| PoET + linear head | regression | 0.573 | 0.515 |
| ProteinNPT (MSAT) | regression | 0.570 | 0.486 |

## 5. Multi-round design on fitness landscapes

### 5.1. Experiment details

We next ask whether the improvements in predictive performance translate to benefits in a multi-round design setting. We follow the evaluation protocol introduced by Notin et al. (2023b) in which design is formulated as a pool-based optimisation task over the sequences in an empirical fitness landscape. For a given landscape, the goal is to retrieve as many high-scoring sequences as possible over the course of 10 optimisation rounds. In each round, the model's predictions are used to guide the selection of a batch of 100 sequences to acquire from a pool of candidate sequences. The pool of candidate sequences is either the complete landscape, or, in the case of the multiple mutant landscapes, a randomly selected subset of 5000 sequences. We follow Notin et al. (2023b) in using ensembling strategies to derive uncertainty estimates which can be used to guide the

selection of candidates from the pool within the framework of Bayesian optimisation (BO). To make the comparison between the modelling strategies as direct as possible, we choose to follow Notin et al. (2023b)'s use of the upper confidence bound (UCB) acquisition function, but note that the use of a ranking loss means that our ensembles should be considered as preferential surrogates within the framework of preferential BO (González et al., 2017), and may benefit from specialised acquisition functions.

We compare optimisation guided by ensembles of PoET and ESM-1v ranking models to ProteinNPT, as well as selected baselines. For ProteinNPT and embedding-augmented baselines, we use Monte Carlo dropout (Gal & Ghahramani, 2016) to produce uncertainty estimates. Models are seeded before the first round with 100 sequences randomly sampled from the landscape. At each round, we rank all remaining sequences in the pool by their acquisition values, and select the top 100 to add to the training set.

We plot the fraction of the top 30% of sequences in the initial candidate pool that are retrieved by the optimisation process as a function of the number of optimisation rounds for both single and multi-mutant in Figure 1. Across both sets of landscapes, the PoET ranking ensemble outperforms all other methods. In general, the design curves show similar trends to the supervised results. Ranking-based fine-tuning outperforms regression-based fine-tuning, and using ensembles leads to the best performance, though a single model also performs very well (Figure 2). While recall of high-fitness sequences saturates for the single mutant landscapes, it improves steadily for the multiple mutant landscapes, since the starting pools are larger, and it is

*Table 3.* Spearman correlations and test set negative log likelihoods of pairwise predictions for single models versus ensembles on the 5 multi-mutant datasets.

| Model name | Loss type | Spearman | | NLL | |
|---|---|---|---|---|---|
| | | $n = 128$ | $n = 512$ | $n = 128$ | $n = 512$ |
| ESM1v | ranking | 0.653 | 0.736 | 1.42 | 0.768 |
| ESM1v ensemble | ranking | 0.677 | 0.753 | 0.841 | 0.584 |
| PoET | ranking | 0.738 | 0.806 | 0.987 | 0.620 |
| PoET ensemble | ranking | **0.752** | **0.818** | **0.750** | **0.507** |

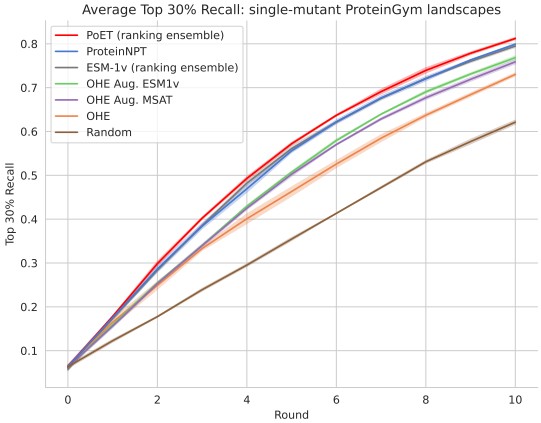
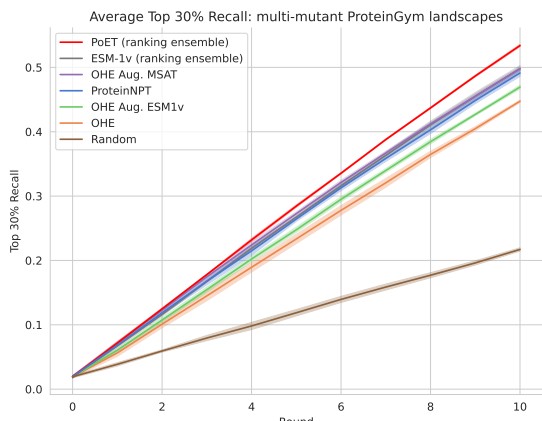

*Figure 1.* Top 30% recall averaged over: (**left**): 8 single-mutant landscapes and (**right**): 5 multi-mutant landscapes. The shading represents one standard deviation over 3 random seeds.

not possible to reach perfect recall within the fixed budget of acquisitions. The relative ordering of the methods is reasonably stable across individual landscapes (Figures 5 and 6), although there are some cases where simple baselines perform comparably to the best-performing methods, suggesting these landscapes may contain noisy or otherwise difficult-to-predict fitness labels (Notin et al., 2023b).

## 6. Conclusion

The ability of language models to learn distributional constraints governing natural protein sequences makes them powerful zero-shot predictors of the effects of mutations. Here we show that their learned distributions can also be rapidly adapted via feedback from relatively few experimental measurements. Even 128 sequences - of the order of a typical batch size in wet lab experiments - allow significant improvements over zero-shot performance. While previous works have also suggested the effectiveness of directly fine-tuning likelihoods, we extend this strategy to the classes of PLM whose distributions best reflect fitness, and find that doing so is crucial to obtaining performance surpassing leading approaches based on frozen embeddings across supervised and multi-round design settings. Notably, fine-

tuning is also dramatically more computationally efficient than the leading embedding-based approaches (Table 4). An intriguing possibility is that when generative PLMs are fine-tuned via likelihood-based loss functions, they may retain their generative capacity, and we believe studying this possibility by leveraging the connection to methods like DPO (Rafailov et al., 2023) to be a promising avenue for future work.

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

## A. Related work

**Zero-shot protein fitness prediction:** The most successful models for zero-shot prediction of protein fitness effects attempt to predict the likelihood of particular sets of mutations occurring within a natural protein given its evolutionary context. Traditional methods within this category involve statistical models trained directly on multiple sequence alignments (MSAs) for each protein of interest, such as profile models (Hopf et al., 2017), Potts models (Figliuzzi et al., 2016; Hopf et al., 2017) and VAEs (Frazer et al., 2021; Riesselman et al., 2018). More recent generalisations of such methods involve pretraining large PLMs across all natural proteins. For example, ESM-1v (Meier et al., 2021) is trained using a masked language modelling objective, allowing point mutations to be scored by the ratio of probabilities of mutant and wild-type amino acids. Alternatively, autoregressive models can directly compute the likelihood of entire protein sequences, making them more appropriate for scoring sequences containing multiple mutations (Notin

et al., 2022; Nijkamp et al., 2023; Madani et al., 2023). However, unconditional PLMs suffer from a lack of context, often requiring fine-tuning to specialise their distributions towards a particular protein family of interest (Madani et al., 2023). As a result, the leading autoregressive models exploit the information in MSAs to improve predictions, either by biasing language model likelihoods with statistics from the MSA, in the case of Tranception (Notin et al., 2022), or by explicitly conditioning on the MSA (Hawkins-Hooker et al., 2021; Ram & Bepler, 2022; Truong Jr & Bepler, 2023).

**Supervised protein fitness prediction:** Fitness prediction has also been studied as a supervised learning task in many prior works (Rao et al., 2019; Hsu et al., 2022a; Krause et al., 2021). Several works have sought to exploit the pre-trained representations of PLMs to improve performance, using either fine-tuned (Rao et al., 2019) or frozen embeddings (Dallago et al., 2021; Notin et al., 2023b). Nonetheless, approaches based on embeddings risk discarding useful distributional information captured in the models' output layers (Krause et al., 2021). The importance of fully leveraging distribution modelling for fitness prediction is highlighted by the success of 'augmented density' predictors (Hsu et al., 2022a), which combine zero-shot fitness predictions with either one-hot encoded (Hsu et al., 2022a), or embedded (Notin et al., 2023b) representations of input sequences. The state-of-the-art supervised fitness prediction method ProteinNPT (Notin et al., 2023b) combines these strategies, training a custom non-parametric Transformer (Kossen et al., 2021) to reason over both zero-shot predictions and associated sequence embeddings to produce fitness predictions.

Methods seeking to adapt the distributions learned by PLMs directly have been less well studied. Rives et al. (2021) proposed to use the log-likelihood ratio between mutant and wild-type amino acids as a regression function, fine-tuning the full model. Krause et al. (2021) suggest using a ranking-based loss function to fine-tune autoregressive PLMs, showing improvements over augmented density baselines on a small set of fitness landscapes. A similar ranking-based loss function was proposed for training non-pretrained CNN architectures on fitness datasets in Brookes et al. (2023). Most recently, Lee et al. (2023) apply ranking-based loss functions derived from the literature on large language model alignment (Rafailov et al., 2023) to fine-tune unconditional autoregressive PLMs. The application of ranking-based loss functions to masked PLMs is also considered in concurrent work (Zhao et al., 2024).

**Model-guided protein design:** Several works have proposed variants of Bayesian optimization (BO) for designing biological sequences, including proteins (Gruver et al., 2021; Jain et al., 2022; Khan et al., 2023; Stanton et al., 2022; Hie & Yang, 2022). The majority of these BO ap-

proaches are evaluated in an unconstrained setting, in which sequences are proposed by the optimiser and evaluated with a black-box oracle designed to mimic a biological property of interest. An alternative *in silico* evaluation strategy avoids the challenge of defining a meaningful oracle function by adopting a pool-based optimisation problem formulation over experimentally determined fitness landscapes (Notin et al., 2023b). Another line of works has sought to provide direct experimental validation of approaches combining uncertainty estimates with PLMs, in settings ranging from zero (Hie et al., 2023) or few-shot design (Biswas et al., 2021) to single-round design given large training sets of sequence-fitness pairs (Li et al., 2023). In this paper, we focus on evaluating different PLM-based fitness prediction strategies in *in silico* settings designed to mimic applied design scenarios. We study both a supervised setting and a model-guided design setting, which extends the pool-based optimisation setting proposed in Notin et al. (2023b), to a set of the most diverse multi-mutant fitness landscapes in ProteinGym.

## B. Fitness landscapes

We use the set of 8 single-mutant landscapes selected for ablations and hyperparameter selection by (Notin et al., 2023b). The names of these landscapes in ProteinGym are:

- BLAT_ECOLX_Jacquier_2013
- CALM1_HUMAN_Weile_2017
- DYR_ECOLI_Thompson_2019
- DLG4_RAT_McLaughlin_2012
- REV_HV1H2_Fernandes_2016
- TAT_HV1BR_Fernandes_2016
- RL40A_YEAST_Roscoe_2013
- P53_HUMAN_Giacomelli_WT_Nutlin

We additionally select a set of 5 of the most diverse multi-mutant landscapes in ProteinGym. To select these landscapes, we identified the landscapes with the largest number of mutations in ProteinGym, and discarded redundant landscapes: for example the GFP landscapes of (Gonzalez Somermeyer et al., 2022) are landscapes of close homologues of the GFP protein whose landscape was reported by Sarkisyan et al. (2016). We therefore include only the latter.

The selected multi-mutant landscapes are:

- PABP_YEAST_Melamed_2013

- CAPSD_AAV2S_Sinai_2021

- GFP_AEQVI_Sarkisyan_2016

- GRB2_HUMAN_Faure_2021

- HIS7_YEAST_Pokusaeva_2019

## C. Decoder-only fine-tuning of PoET

PoET parameterises a sequence of conditional distributions over the amino acids in a set of protein sequences in the same family. The model represents the joint likelihood of a set of sequences $M = \{m^{(1)}, ..., m^{(N)}\}$, via an autoregressive factorisation over sequences and over positions within each sequence:

$$p(M) = \prod_i p(m^{(i)} | m^{(<i)}) = \prod_{ij} p(m_j^{(i)} | m_{<j}^{(i)}, m^{(<i)}).$$
(9)

To parameterise this distribution, PoET uses a causally masked Transformer architecture, which maps from previous amino acids to logits for the current amino acid. Conceptually, this function can be decomposed into two stages: first the entire history of previous sequences $m_{<i}$ is encoded into a sequence of embeddings $H_{<i} \in \mathbb{R}^{L_{<i} \times D \times E}$, where $D$ is the number of layers and $E$ is the embedding dimension, via a stack of causally masked layers:

$$H_{<i} = f_\theta(m^{(<i)}).$$
(10)

The current sequence $m_i$ is then decoded by a function which maps these prior sequence embeddings and previous amino acids in the current sequence to logits for each position $j$:

$$\text{logit}_{ij} = g_\theta(m_{<j}^{(i)}, H_{<i}).$$
(11)

To fine-tune PoET from fitness data, we propose to fine-tune only the weights of the function $g$, representing the 'decoding' of the current sequence given its context. To achieve this, we first clone the PoET weights, producing a set of 'encoder' weights $\phi$ and a set of 'decoder' weights $\theta$. We use the frozen encoder weights to produce an embedding $H \in \mathbb{R}^{L_M \times D \times E}$ of the input MSA sequences: $H = f_\phi(\{m^{(1)}, ..., m^{(N)}\})$, where $L_M$ is the total length of all sequences in the input MSA. We then fine-tune the weights $\theta$ of the cloned 'decoder' to minimise the cross-entropy loss of Equation 2 on the labelled data. Concretely, the scoring function used to parameterise the Bradley-Terry model becomes:

$$s_\theta(x, M) \equiv s_\theta(x, H) = \sum_i \log p_\theta(x_i | x_{<i}, H)$$
(12)

To maximise computational efficiency, the MSA embeddings $H$ are pre-computed before the start of the fine-tuning process, and remain frozen throughout.

## D. Hyperparameter details

Hyperparameters for the fine-tuning methods are selected based on performance on the single mutant set, consistent with the practice used to select hyperparameters for the baselines from ProteinNPT. We report metrics obtained when using these hyperparameters on both single-mutant and multiple-mutant landscapes for each method.

ESM-1v, ProGen2 and PoET models were fine-tuned using the Adam optimizer (Kingma & Ba, 2015) using gradient accumulation with an effective batch size of 32. We use the first of the five ESM-1v checkpoints. Learning rates for regression-based and ranking-based fine-tuning were selected separately in each case after after a sweep over the values $1e-4, 3e-5, 1e-5$ on the 8 single mutant landscapes. For ESM-1v, we computed the loss by scoring all sequences using the logits generated by passing the wild-type sequence through the model in a single forward pass. In the fitness prediction experiments, the models were trained for 50 epochs. During training on each landscape the Spearman correlation on a separate validation set of 128 sequences from the landscape was used to determine the epoch whose checkpoint should be used to produce predictions on the test set.

### D.1. Regression heads

Linear regression heads were added to embeddings extracted from PoET and ESM-1v. In the former case, we used final token embeddings, and in the latter case we averaged embeddings across the sequence dimension before feeding them to the regression head.

### D.2. Ensembles

Ensembles of size 5 were used for both ESM-1v and PoET. During design, the ensemble members were trained for a fixed number of epochs (15 for PoET; 20 for ESM-1v) each round. All ensemble members were reinitialised from the pretrained model each round.

## E. PoET MSA subsampling

For PoET, in both single-model and ensemble configurations, we sampled context sequences from the same filtered MSAs used to extract MSA Transformer embeddings for

ProteinNPT. These MSAs are generated from the full MSAs provided with ProteinGym by running hhfilter, requiring a minimum coverage of 75% and a maximum sequence identity of 90%. Subsequently, we use weighted sampling to select sequences to pass as context to PoET, up to a maximum context length of 8192 tokens. The MSA is encoded using a frozen copy of the PoET model into a set of cached hidden representations, as described in Appendix C. When ensembling, a separate MSA is sampled for each ensemble member, and held fixed during the fine-tuning of that ensemble member.

## F. Baseline models

ProteinNPT, the embeddings augmented (Emb. aug.) baselines, and the one-hot encoding augmented (OHE aug.) baselines, were all run using the code released by (Notin et al., 2023b). The one-hot and embedding augmented models both use the strategy from (Hsu et al., 2022a) of combining the zero-shot predictions from a pretrained model with sequence features in a regression framework. They differ in the way sequence features are extracted: in the former case, ridge regression is performed directly on the one-hot encoded sequences. In the latter case, PLM embeddings are used to featurise the sequences. We refer to (Notin et al., 2023b) for further details.

For the fitness prediction experiments, separate ProteinNPT models were trained for 2000 and 10000 steps, and the results of the best-performing model were reported. The other baselines appeared to benefit more from longer training and were trained for 10000 steps, as in (Notin et al., 2023b). For design experiments, we used the Monte Carlo dropout uncertainty quantification strategy proposed by (Notin et al., 2023b) for both ProteinNPT and baselines. Notin et al. (2023b) report best results with a 'hybrid' uncertainty quantification strategy, however this strategy is not implemented in the publicly available code.

## G. Compute requirements

All experiments were run on either V100 or A100 NVIDIA GPUs. Compute required for a single fine-tuning run varies based on the model, the length of the protein sequences, and the size of the dataset. We provide representative timings for the AAV dataset in Table 4. Design experiments involved 10 rounds of fine-tuning and therefore required roughly ten times the computation of a single fine-tuning run.

## H. Additional design plots

We compare different PoET configurations for design on the multiple mutants landscapes in Figure 2. We provide per-landscape plots at the end of the Appendix.

*Table 4.* Representative run times for fine-tuning on the AAV landscape ($n = 512$) on an A100 GPU, averaged across 3 seeds.

| Model name | Time |
| --- | --- |
| ProteinNPT (MSAT) | 4h 40 m |
| ESM1v regression | 2h 27 m |
| ESM1v ranking | 4 m |
| PoET ranking | 41 m |
| PoET regression head | 36 m |

## I. Performance by landscape for supervised experiments

We provide barplots summarising per-landscape performance for selected models on the $n = 128$ single and multi-mutant splits in Figures 2 and 3.

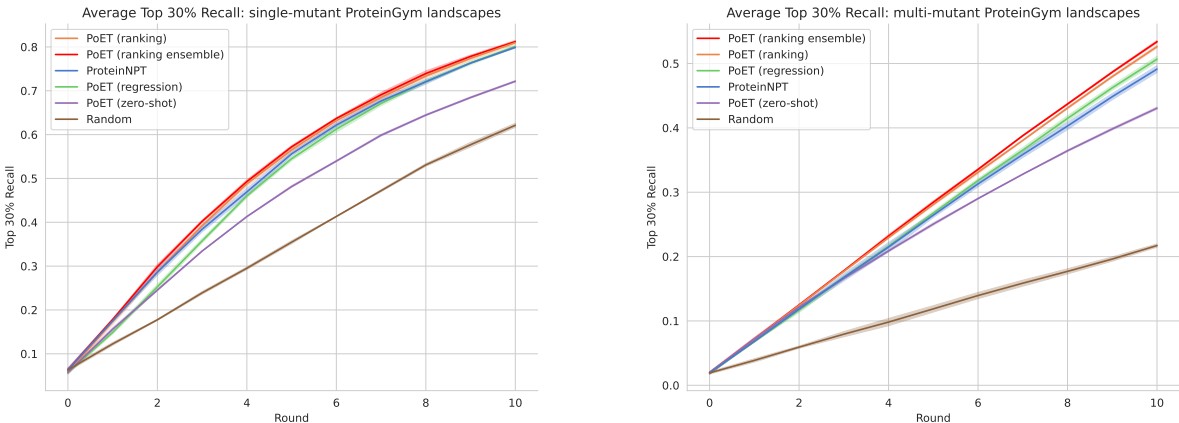

*Figure 2.* **Left**: Average top 30% recall for 8 single-mutant landscapes for alternative PoET configurations as well as selected baselines. **Right**: Average top 30% recall for 5 multi-mutant landscapes for alternative PoET configurations as well as selected baselines.

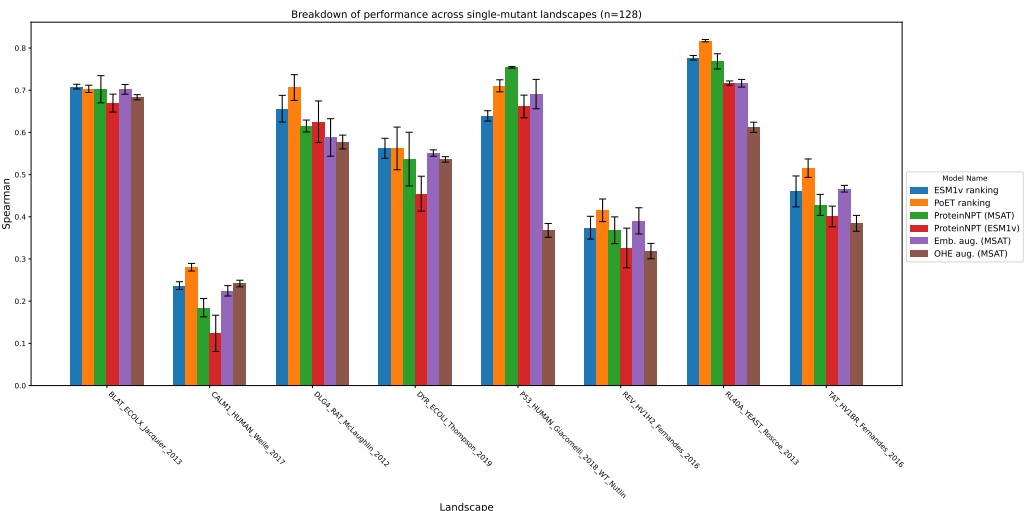

*Figure 3.* Per-landscape performance for singles datasets ($n = 128$). Error bars represent standard deviations across the three train/test splits. The shading represents one standard deviation over 3 random seeds.

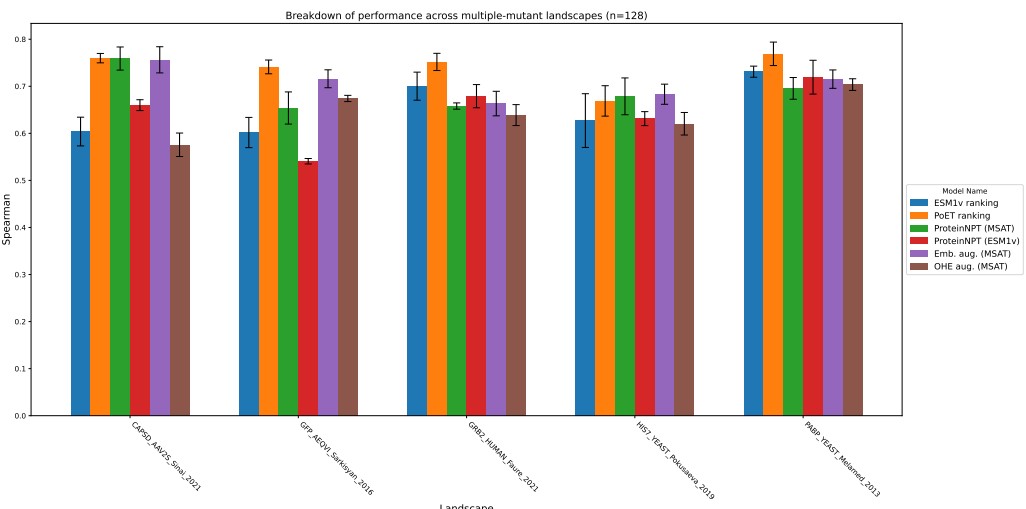

*Figure 4.* Per-landscape performance for multiples datasets ($n = 128$). Error bars represent standard deviations across the three train/test splits.

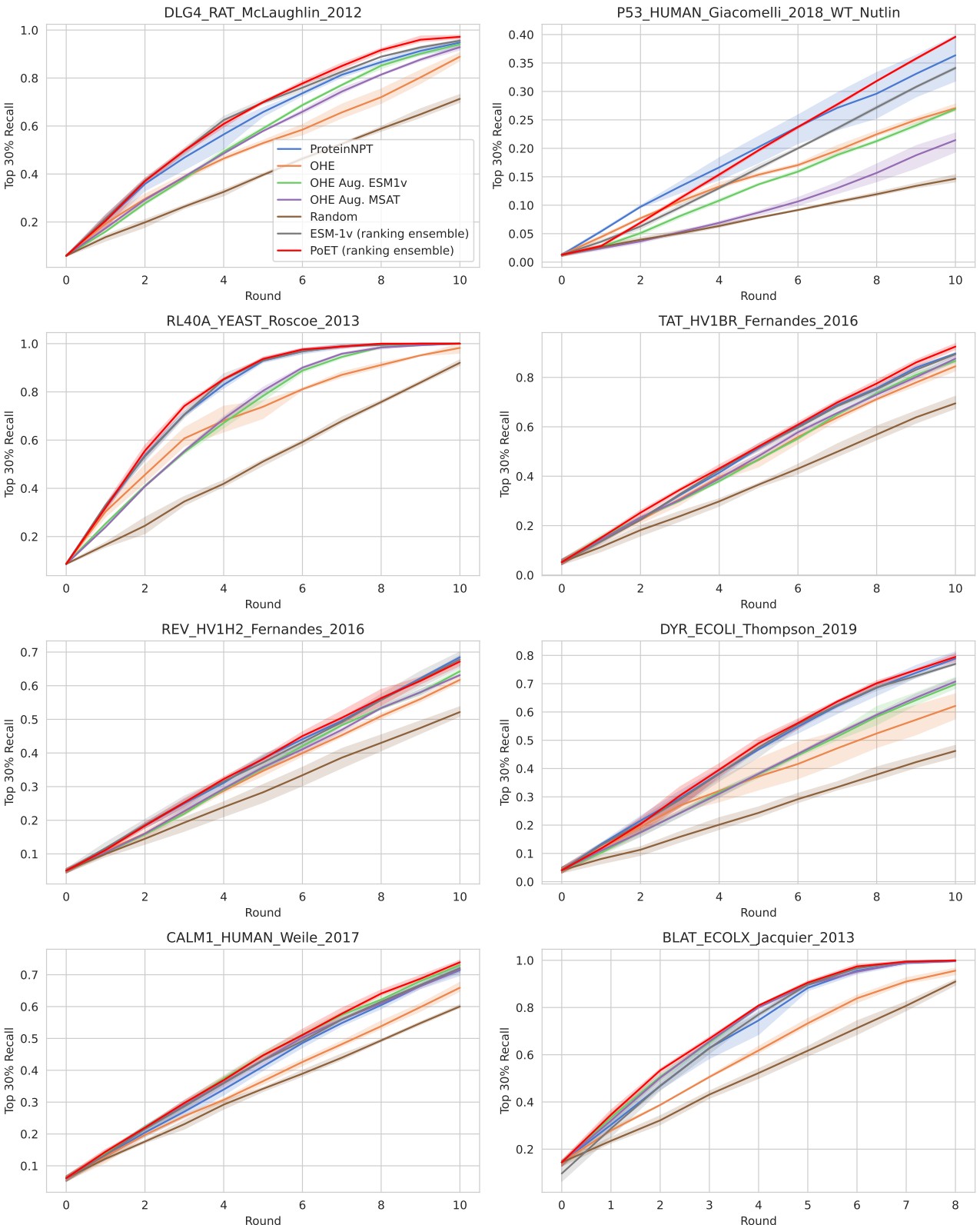

*Figure 5.* Design curves for individual single mutant landscapes.

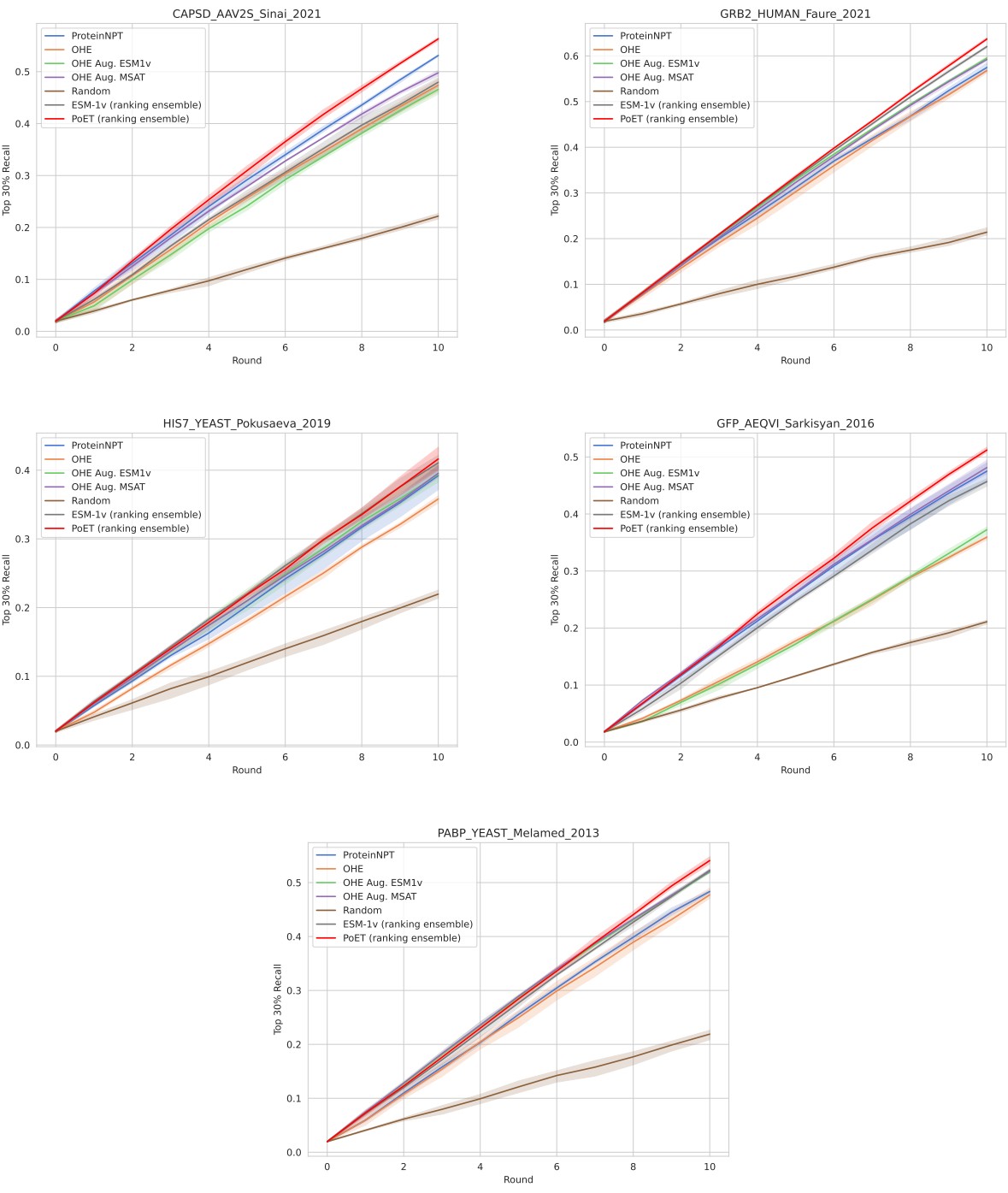

*Figure 6.* Design curves for individual multi-mutant landscapes