# OpenReview forum: "Likelihood-based fine-tuning of protein language models for few-shot fitness prediction and design"
_ICML.cc/2024/Workshop/ML4LMS — ML4LMS Poster_

### Official Review · Reviewer_LUx4 · 2024-06-10
**Interesting paper that evaluates applying ranking losses to fine tune protein language models for few-shot fitness prediction**

**Rating:** 7
**Confidence:** 4

**Review:**

Summary: The manuscript proposes and evaluates ranking losses for fine-tuning protein language models (pLM) for few-shot fitness prediction. They compare ranking-based and regression-based approaches, while also comparing against frozen pLM embeddings. Additionally, they also propose an interesting ensemble strategy combining multiple fine-tuned pLMs.

Additional Comments:

1. The results obtained by the authors are in agreement with recent works in the field. The reviewer suggests checking the work by Lee et al, "Fine-tuning protein Language Models by ranking protein fitness" (2023).
2. The section "Uncertainty quantification with evolutionary context ensembles" shows a promising and interesting research direction. It would be interesting to evaluate how it also can be adapted for different sequence design approaches.
3. For a lab-in-the-loop setting, it would be interesting to have more explanation and a deeper analysis discussing the different forms on how these methods can be applied in a multi-round setting.

---

### Official Review · Reviewer_Luzq · 2024-06-11

**Rating:** 9
**Confidence:** 4

**Review:**

$\textbf{Summary:}$ The paper presents a comprehensive study on the fine-tuning of PLMs for predicting protein fitness and guiding protein design. The authors extend ranking-based loss functions to adapt the likelihoods of both masked and family-based PLMs. They demonstrate that these fine-tuned models outperform state-of-the-art approaches that rely on frozen embeddings, particularly in low-resource settings. The paper also explores ensembling strategies to enhance the performance and uncertainty quantification of these models.

$\textbf{Overall Assessment:}$ The paper is well-motivated, methodologically sound, and contributes valuable insights to the field of protein fitness prediction and design. The experiments are well-aligned with the objectives, and the results demonstrate clear improvements over existing methods. Given the strengths in motivation, flow, and experimental design, I am fairly confident that this paper is among the top submissions at the workshop and is already publication-ready in its current form. It adds significant value to the community by advancing few-shot learning for protein design and offering practical strategies for leveraging PLMs in low-resource data scenarios.

$\textbf{Strengths:}$

1. The motivation of the paper is highly relevant as it addresses a critical challenge in protein engineering - making accurate predictions with limited sequence-function experimental data.
2. The extension of ranking-based loss functions to both masked and family-based PLMs represents a significant methodological advancement.
3. The authors conduct experiments on mutational landscapes from ProteinGym across multiple random splits, using both single-mutant and multi-mutant datasets. ProteinGym is one of the major benchmarks for protein fitness prediction. Hence, this thorough evaluation strengthens the validity of their findings.
4. By comparing their methods against state-of-the-art baselines, including regression-based fine-tuning and models operating on frozen embeddings, the authors provide clear evidence of the superiority of their approach.
5. The proposed ensembling strategies, which leverage the dependence of mutational distributions on sequence context, are a novel contribution and are shown to enhance model performance in the paper.

$\textbf{Potential Areas for Improvements and Suggestions:}$
While I am nitpicking at this point, the paper would be further strengthened by evaluating the fine-tuning strategies on various splits of mutational landscapes from FLIP[1] and additional fitness/activity prediction tasks from PEER[2].

$\textbf{References:}$

[1] Dallago, Christian, et al. "FLIP: Benchmark tasks in fitness landscape inference for proteins." bioRxiv (2021): 2021-11.

[2] Xu, Minghao, et al. "Peer: a comprehensive and multi-task benchmark for protein sequence understanding." Advances in Neural Information Processing Systems 35 (2022): 35156-35173.